# Diversity, Economic Value and Regional Distribution of Plant Food Products at Local Tropical Markets: A Samoa Case Study

**Vladimir Verner [1], Pavel Novy [2], Jan Tauchen [2], Lukas Huml [3], Julian Wong Soon [4], Tomas Kudera [5], Fiti Laupua [4] and Ladislav Kokoska [5,\*]**

[1] Department of Economics and Development, Faculty of Tropical AgriSciences, Czech University of Life Sciences Prague, Kamycka 129, 165 21 Prague 6-Suchdol, Czech Republic; vernerv@ftz.czu.cz

[2] Department of Quality of Agricultural Products, Faculty of Agrobiology, Food and Natural Resources, Czech University of Life Sciences Prague, Kamycka 129, 165 21 Prague 6-Suchdol, Czech Republic; novy@af.czu.cz (P.N.); tauchen@af.czu.cz (J.T.)

[3] Department of Chemistry of Natural Compounds, University of Chemistry and Technology Prague, Technicka 5, 166 28 Prague 6–Dejvice, Czech Republic; lukas.huml@vscht.cz

[4] Scientific Research Organisation of Samoa, P.O. Box 6597 Apia, Western Samoa; wongsjuli@chem.eng.osaka-u.ac.jp (J.W.S.); fiti.laupua@sros.org.ws (F.L.)

[5] Department of Crop Sciences and Agroforestry, Faculty of Tropical AgriSciences, Czech University of Life Sciences Prague, Kamycka 129, 165 21 Prague 6-Suchdol, Czech Republic; kuderat@ftz.czu.cz

\* Correspondence: kokoska@ftz.czu.cz; Tel.: +420-224-382-180

**Abstract:** Local markets are still an integral part of the food system in developing economies of tropical regions including Samoa. This small South Pacific country is largely dependent on the production of crops in village agriculture, where traditional markets play an important role in sustainability of food supply. Similarly as many small island developing economies, Samoa is currently facing several challenges such as food security and high dependence on food imports. Therefore, we decided to monitor the diversity of plant foods on Samoan local markets and their economic and geographic indicators through interviews with the vendors. Our results suggest that assortment and economical value of plant food products have potential to increase sustainable food security of the local population and support economic growth of the region. For example, underutilized crops available at local markets are prospective species for development of new food products with beneficial nutritional and health properties. Moreover, certain commodities (e.g., papaya, kava and Samoan cocoa) seem to be promising for export. In addition, our findings suggest that development of appropriate processing technologies and the optimization of the logistics of crop products sold at local markets can contribute to an increase in efficiency of the regional agricultural sector.

**Keywords:** Samoa; local market; inventory; cash crop; agricultural production

---

## 1. Introduction

A local market might be described as a location frequented by clients and customers who regularly gather and sell products in the region or area in which they are produced. In comparison with large industrialized markets, they have specific production, processing and distribution patterns. For example, local agricultural produce markets are characterized by regional food commodities produced under sustainable agriculture that are distributed over short distances and sold directly by farmers to consumers [1,2]. It is important to mention that despite the continuously increasing role of supermarkets on consumers' dietary intake and habits in the tropical regions, local markets

still play an important role in supplying population dietary needs due to their long tradition, accessibility and the fact that the majority of food products are perishable and must be consumed soon after harvesting [3]. Since the local agricultural produce markets form part of the local food system practices, they significantly contribute to food and health security in both developed and developing countries [4,5]. Typically, they are based on locally available resources having improved economic viability to both farmers and consumers. Their production and distribution practices are more environmentally friendly in comparison to industrialized food systems and they enhance social equity for all members of the particular community [6–8]. Additionally, local food systems usually have better availability of fresh items and thus might provide health benefits due to their superior nutritional quality [9,10]. Since the local food systems address the environmental, nutritional, health, economic and social well-being of a specific urban or rural population, studying and gaining a deeper knowledge of local markets have received much interest over the past few years [2].

Market surveys documenting locally sold plant food items are cost-effective techniques providing qualitative and quantitative data concerning the importance, preferences and value that local population imputes to the different edible species [11,12]. Moreover, surveys might give a picture if people have moved away from a culture-based traditional use of the surrounding plants to one grounded on imported goods [13]. An analysis of three main factors might give a good overview of the local market situation and the general habits of local communities: (i) the economic value of given product, (ii) the richness of assortment and (iii) the regional distribution of a particular product [14]. All of these above-mentioned aspects are very important for implementation of socioeconomic improvement strategies aimed at generating economic growth, food security and the conservation of local plant diversity. Besides social and health benefits, economic data (particularly price, supply frequencies and daily sold volumes) of locally sold items might give interesting information about the overall situation of demand and supply and the availability of particular products [15,16]. Moreover, assortment depth and width reflect the richness of diversity in a given location and the proportion of locally-produced/versus imported foods [17,18]. Finally, regional distribution gives an overview of the operation of local market chains, i.e., where agricultural commodities were produced and from what distance they have been transported to their vending location. This information is useful for any understanding of existing market chains as it is necessary to know by whom the particular product is sold to consumers; how possible extra transportation expenditures affects the final price and its environmental-friendliness and what knowledge of the product is shared between vendor and consumer [2]. Data obtained from surveys of local markets, which are still an integral part of the food system in developing economies of tropical regions, are valuable for the identification of new crops/food products [19]. A majority of market surveys performed in tropical countries are therefore chiefly focused on gathering ethnobotanical knowledge [20–22]. Despite the existence of sporadic studies describing the economic situation of locally produced plant foods in tropical regions such as Central Africa, South and South-East Asia [12,23–25], the research focused on an advanced economic analysis of local food markets is almost exclusively restricted to industrialized countries [4,26].

The Independent State of Samoa (formerly known as Western Samoa) is a small South Pacific island country belonging to the Polynesian region whose economy is historically dependent on the production of cash crops in village agriculture [27]. Here, traditional markets still represent a very important component of the local food chain system. However, the current numbers show that the Samoan market is insufficiently supplied with certain crop products and Samoans are becoming more dependent on imported plant foods [28]. The current critical situation can be illustrated by official numbers showing that the agricultural area of Samoa declined from 58,000 to 22,000 ha under permanent crops and from 18,000 to 8000 ha of arable land in 1988 and 2015, respectively [29]. In addition, it was previously documented that the local population had experienced a shift from a culture based on the traditional use of the surrounding plants to one based on foreign food resources. As a result of this process, many wild and cultural plants have fallen into disuse and have been forgotten, whereas some species have become extinct since they were no longer needed or grown [30]. These changes represent

a serious risk to the future sustainable supply of Samoans with affordable healthy foods. Indeed, the inventory of Whistler [13] documented plant foods sold in Samoan market places to be almost entirely composed of commonly known tropical and subtropical crops. This is in contrast with the fact that the Samoan Islands are considered a biodiversity hotspot and genetic centre of Polynesian plant species, harboring approximately 550 vascular plants, of which about 30% are endemic [30]. Correspondingly, the diversity of plant products at Samoan local markets represents a promising source for the identification of new species of underutilized crops [13]. Although the Samoa Bureau of Statistics (SBS) maps on a regular basis general economic data on common/major crops being sold at local markets, detailed analysis of the assortment, economic value and regional distribution of plant food products at local markets in Samoa is still lacking.

In view of the above-mentioned facts, we decided to monitor the diversity of plant foods on Samoan markets and their economic and geographic indicators through interviews with the vendors with the aim to illustrate the economic potential of plant food products at local tropical markets. The specific goals were (i) to document the diversity of commercialized plant species, (ii) to determine the stock volumes and financial values of the main cash crops (iii) and to map the regional distribution of their production.

## 2. Materials and Methods

### 2.1. Study Area

The data were chiefly collected in the local markets situated along the northern and eastern coasts of Upolu and Savai'i Islands, respectively. In Upolu, where the majority of local markets are concentrated, the interviews with vendors were performed in the capital city Apia at Fugalei, Lemoasina and Taufusi markets, on the main road west from Apia in the town Vaitele (Vaitele market) and the village Afega (Laumua o Tumua market). Fugalei, the biggest farmers market in the country located close to the center of Apia, offers various fresh produce, traditional Samoan foods and handcrafts. Lemoasina is a small market consisting of several simple stands localized just opposite the southeast corner of the Fugalei market. Taufusi is a drive through market with covered stands; located about 200 m south from Fugalei. The relatively modern infrastructure of the public market in Vaitele, opened in 2011, is poorly utilized, probably due to the close vicinity of Apia city. Laumua o Tumua is a new local community market located on the northwestern coast of Upolu, opened in 2014. Regarding Savai'i Island, the data was gathered in Salelologa market, which is a relatively huge marketplace near Salelologa harbor. It is the only official market on the whole island. For larger markets (e.g., in Vaitele and Salelologa), Samoa Land Corporation defines finance policy and guidelines for stock control of commodities sold there. The prices of the products are shown by vendors in Samoan tālā (WST), but a bargain is sometimes possible. The markets have official opening and closing times. Since the individual stands along the roads where farmers sell their agricultural produce are also relatively abundant throughout both of the islands, data was also obtained from several individual vendors selling their products usually in front of local shops in Muiatele village (Upolu), Salelologa city and the villages of Falealupo and Safotu (Savai'i). The stands scattered all around the islands in front of houses along the main roads were not included. These stands usually have on display just one or two products, the most frequent being coconuts.

## 2.2. Data Collection and Analysis

The survey was conducted in July and August 2014, from August to October 2015 and from May to June 2018 via face-to-face interviews using a structured questionnaire in English, with all vendors willing to participate interviewed. After disclosing fully the intent and scope of the research, the prior informed consent agreement was obtained by the researcher from the vendor. Then, sociodemographic characteristics of the vendors, namely age, gender, place of residence and occupation, were recorded. The assortment of crop species available at each particular stand was identified and local names of crops were verified with data from the literature [13]. Subsequently, the origin of the produce, price, stock and quantities sold were recorded. An average weight of sales unit was determined using a laboratory balance (SSH91, Scaltec Instruments, Gottingen, Germany) and annual volumes for sale and price per kilogram were calculated following methods adapted from similar studies [31,32]. Since the quantitative data on the main cash crops was collected in 2014, the prices of the monitored plant foods recorded in WST were converted to United States dollars (USD) according to the official average 2014 exchange rate (USD 1 = WST 2.33) of The World Bank [33]. For comparison of data collected, all results are expressed in the text as a percentage, whereas stock volume and the financial value of the main cash crops is related to the number of all vendors interviewed ($n$ = 208) and the regional distribution of the vendors, the diversity and market value of the cash crops to the farmers selling their own produce only ($n$ = 148). Crop categories of plant species were classified according to the literature data [13]. Neglected/underutilized crops were categorized based on characteristics proposed by International Plant Genetic Resources Institute [34]. Professor Kokoska verified the authenticity of commonly known agricultural crops and identified neglected crop species, and their voucher specimens are deposited in the herbarium of the Department of Botany and Plant Physiology of the Faculty of Agrobiology, Food and Natural Resources of the Czech University of Life Sciences Prague, Czech Republic. The scientific names of all species were also reviewed using The Plant List [35].

## 3. Results

### 3.1. Demographic Characteristics of Vendors

Information was obtained from a total of 208 vendors, with 63.5% and 36.5% of them selling their products in Upolu and Savai´i Islands, respectively. Out of the total number of vendors questioned, 41.8% were male and 58.2% were female. Their age ranged from 13 to 70 years with an average age of 37 and 42 for men and women, respectively. For 88.5% of respondents market vending was their main occupation, whereas 11.5% had other profitable occupation excluding laboring on their own plantations. Typical examples of other vendors' occupations were bartender, cook, manual worker, officer and teacher. Most of them (61.5%, $n$ = 128) sell their products on a daily basis, except Sunday. The majority (71.2%) of the vendors were selling their own produce. The data on demographic characteristics of vendors is summarized in Table 1.

**Table 1.** Demographic data on respondents.

| Variables | Number of Respondents (*n* = 208) |
|:---:|:---:|
| Gender | |
| Male | 87 |
| Female | 121 |
| Age (years) | |
| ≤17 | 5 |
| 18–30 | 47 |
| 31–60 | 142 |
| ≥61 | 14 |
| Region of residence | |
| Upolu | 132 |
| Savaiʻi | 76 |
| Occupation | |
| Vendor/farmer | 184 |
| Other | 24 |

*3.2. Diversity of Commercialized Plants*

During our survey, 53 plant species, their varieties, genetic groups and subgroups belonging to 30 families and 6 categories of agricultural crops were recorded. Musaceae and Solanaceae were the most diverse families (each represented by 9.4% of plant species), followed by Cucurbitaceae (7.5%), Araceae, Brassicaceae, Leguminosae, Rutaceae (5.7% each), Dioscoreaceae, Moraceae, Rubiaceae and Zingiberaceae (3.8% each). The remaining families were only composed of one species. Fruits and nuts (45.3%) were the highest represented agricultural crops, followed by vegetables (24.5%), tuber crops (15.1%), stimulants (7.5%), spices (3.8%), algae and cereals (1.9% each). Correspondingly, the most frequently used plant parts were fruits (including infrutescences) and seeds (66.0%), rhizomes, roots and tubers (22.6%) and leaves (7.5%). Documented species were predominantly perennial herbs and trees (32.1% each), followed by annual herbs (13.2%), biennial herbs and perennial vines (7.5% each). Only one representative was recorded each for algae, annual vines, palms and shrubs. Higher crop diversity was found in Upolu (55.2%) than in Savaiʻi (44.8%). The vast majority of the species identified during our study in the market were common agricultural crops (90.6%); however, certain plants, namely *Adenanthera pavonina*, *Inocarpus fagifer*, *Musa troglodytarum* and *Tacca leontopetaloides* were identified as neglected/underutilized crops, having importance only on a local level. The local variety of banana (*Musa × paradisiaca*; group AAB) called faʻi sāmoa (Samoan banana) can also be included in the same category. Nonseasonal, seasonal and imported (not cultivated in Samoa) crops accounted for 77.4%, 20.8% and 1.9% of the total number of plant species sold in the markets, respectively. The most crops were of moderate availability (58.5%) and cultural salience (69.8%). Largely available (22.6%) and highly culturally important (18.9) plants were dominant over the scarce (18.9%) and low salient (11.3%) species. The detailed data on plant species identified in Samoan markets including their category of use, life form, part used, locality, crop category, family, seasonality, availability, traditional uses and cultural salience and their scientific, English and local names are listed in Table 2.

**Table 2.** Assortment of plant species on Samoan markets.

| Scientific Name (VSN [1]) | Family | English Name | Local Name | Life Form | Part Used | L [2] | CC [3] | Seasonality/Availability | Traditional Uses/Cultural Salience |
|---|---|---|---|---|---|---|---|---|---|
| | | | | | Cereals | | | | |
| *Zea mays* L. | Poaceae | corn | sana | annual herb | seed | U | C | nonseasonal/scarce | boiled and roasted/low |
| | | | | | Root and tuber crops | | | | |
| *Alocasia macrorrhizos* (L.) G.Don | Araceae | giant taro | ta'amu | perennial herb | rhizome | U, S | C | nonseasonal/large | baked in umu or boiled with coconut cream, sometimes baked wrapped in breadfruit or banana leaves, and with coconut cream/high |
| *Colocasia esculenta* (L.) Schott | Araceae | taro | talo | perennial herb | rhizome | U, S | C | nonseasonal/large | baked in umu, boiled with coconut cream, grated and baked then eaten with caramelized sugar mixed in coconut cream, deep fried sliced thin pieces (taro chips)/high |
| *Dioscorea alata* L. | Dioscoreaceae | yam | ufi | perennial vine | tuber | U, S | C | seasonal (October-December)/moderate | baked in umu, sometimes wrapped in breadfruit or banana leaves or boiled with coconut cream/moderate |
| *Ipomoea batatas* (L.) Lam. | Convolvulaceae | sweet potato | ´umala | perennial vine | tuber | U | C | nonseasonal/moderate | baked in umu, sometimes wrapped in breadfruit or banana leaves, and with coconut cream/moderate |
| *Manihot esculenta* Crantz | Euphorbiaceae | cassava | manioka | perennial herb | tuber | U, S | C | nonseasonal/moderate | boiled in coconut cream, cooked pieces with warm sweet coconut cream, starch production/moderate |
| *Solanum tuberosum* L. | Solanaceae | potato | pakeka | perennial herb | tuber | U, S | C | not cultivated (imported)/scarce | salad from cooked tubers cut into cubes mixed with vegetables and spices/low |
| *Tacca leontopetaloides* (L.) Kuntze (2440KBFR) | Dioscoreaceae | Polynesian arrowroot | māsoā | perennial herb | tuber | U, S | N | nonseasonal/scarce | fried after mixing with water and eaten with caramelized sugar, starch extracted from tuber is added to other foods/moderate |
| *Xanthosoma sagittifolium* (L.) Schott | Araceae | tannia | talo palagi | perennial herb | rhizome | S | C | nonseasonal/moderate | baked in umu, sometimes wrapped in breadfruit or banana leaves, and with coconut cream/low |
| | | | | | Fruits and nuts | | | | |
| *Adenanthera pavonina* L. (03014KBFR) | Leguminosae | red bead | lopa | tree | seed | U, S | N | seasonal (December)/scarce | eaten roasted/moderate |
| *Ananas comosus* (L.) Merr. | Bromeliaceae | pineapple | fala ´aina | perennial herb | infrutescence | U, S | C | seasonal (November–December)/moderate | eaten as fresh fruit, used as an ingredient in many dishes and drinks such as fruit salads, pies and juices/moderate |
| *Annona muricata* L. | Annonaceae | soursop | sasalapa | tree | fruit | U | C | nonseasonal/moderate | eaten as fresh fruit and used as an ingredient for fruit salads and juices/moderate |
| *Artocarpus altilis* (Parkinson ex F.A.Zorn) Fosberg | Moraceae | breadfruit | ´ulu | tree | infrutescence | U | C | seasonal (June and December)/moderate | baked in umu or boiled, with coconut cream, roasted and mashed eaten with caramelized sugar in coconut cream, used as an ingredient in soups/high |

**Table 2.** *Cont.*

| Scientific Name (VSN [1]) | Family | English Name | Local Name | Life Form | Part Used | L [2] | CC [3] | Seasonality/Availability | Traditional Uses/Cultural Salience |
|---|---|---|---|---|---|---|---|---|---|
| *Artocarpus heterophyllus* Lam. | Moraceae | jackfruit | ´ulu initia | tree | infrutescence | U | C | nonseasonal/moderate | eaten as fresh fruit/moderate |
| *Averrhoa carambola* L. | Oxalidaceae | star fruit | vineta | tree | fruit | U | C | nonseasonal/moderate | sliced and served in salads, use to add flavor and as a garnish on foods and drinks/moderate |
| *Carica papaya* L. | Caricaceae | papaya | esi | perennial herb | fruit | U, S | C | nonseasonal/large | eaten as fresh fruit, cooked with coconut cream, cocoa and starch/moderate |
| *Citrus aurantiifolia* (Christm.) Swingle | Rutaceae | lime | típolo | tree | fruit | U, S | C | nonseasonal/moderate | use to add flavor and as a garnish on foods and drinks/moderate |
| *Citrus × aurantium* L. | Rutaceae | bitter orange | moli ´aina | tree | fruit | U, S | C | seasonal (December)/moderate | eaten as fresh fruit, drunk as a juice/moderate |
| *Citrus maxima* (Burm.) Merr. | Rutaceae | pomelo | moli meleke | tree | fruit | U | C | nonseasonal/moderate | eaten as fresh fruit, drunk as a juice/moderate |
| *Cocos nucifera* L. | Arecaceae | coconut | niu, popo | palm | fruit | U, S | C | nonseasonal/large | flesh and water from green fruits are eaten and drunk, coconut cream is used in many traditional foods, oil is used in frying/high |
| *Inocarpus fagifer* (Parkinson) Fosberg (03015KBFR) | Leguminosae | Tahitian chestnut | ifi | tree | seed | S | N | seasonal (December-January, June-July)/scarce | cooked by placing the unhusked fruits on the fire/moderate |
| *Mangifera indica* L. | Anacardiaceae | mango | mago | tree | fruit | U, S | C | seasonal (November)/moderate | eaten as fresh fruit, either unripe or ripe/moderate |
| *Morinda citrifolia* L. | Rubiaceae | Indian mulberry | nonu | tree | fruit | S | C | nonseasonal/moderate | eaten as fresh fruit or cooked/moderate |
| *Musa acuminata* Colla (sub-group AAA) | Musaceae | Cavendish banana | fa´i pālagi | perennial herb | fruit | U, S | C | nonseasonal/large | eaten as fresh fruit, drink prepared by mashing ripe bananas in coconut cream with limes or lemon grass/high |
| *Musa × paradisiaca* L. (group AB) | Musaceae | lady finger banana | misiluki | perennial herb | fruit | U, S | C | nonseasonal/large | eaten as fresh fruit/high |
| *Musa × paradisiaca* L. (group AAB) | Musaceae | plantain | fa´i pata | perennial herb | fruit | U, S | C | nonseasonal/large | baked in umu wrapped in banana leaves with coconut cream, whole or mashed boiled in water or coconut cream/high |
| *Musa × paradisiaca* L. (group AAB) [04012KBFR] | Musaceae | Samoan banana | fa´i sāmoa | perennial herb | fruit | U, S | N | nonseasonal/moderate | eaten as fresh fruit/high |
| *Musa troglodytarum* L. (04013KBFR) | Musaceae | mountain plantain | soa´a | perennial herb | fruit | U | N | nonseasonal/scarce | baked or boiled in coconut cream/moderate |
| *Nephelium lappaceum* L. | Sapindaceae | rambutan | fulufulu | tree | fruit | U | C | seasonal (June)/scarce | eaten as fresh fruit/low |
| *Passiflora edulis* Sims | Passifloraceae | passionfruit | pásio | perennial vine | fruit | U | C | nonseasonal/moderate | eaten as fresh fruit, used as an ingredient in drinks such as fruit juices/moderate |
| *Persea americana* Mill. | Lauraceae | avocado | ´āvoka | tree | fruit | U, S | C | seasonal (November-December)/moderate | eaten raw and used in salad/moderate |
| *Pouteria caimito* (Ruiz and Pav.) Radlk. | Sapotaceae | abiu | polo o atamu | tree | fruit | U | C | nonseasonal/scarce | eaten as fresh fruit/low |
| *Syzygium malaccense* (L.) Merr. and L.M.Perry | Myrtaceae | Malay apple | nonu fi´afi´a | tree | fruit | S | C | seasonal (November-February)/moderate | eaten as fresh fruit/moderate |
| | | | | | | Vegetables | | | |
| *Allium cepa* L. | Amaryllidaceae | onion | aniani | biennial herb | leaf | U, S | C | nonseasonal/moderate | ingredient used for salads and cooking/moderate |
| *Brassica rapa* L. cv. group Pack Choi | Brassicaceae | Chinese cabbage | kapisi saina | biennial herb | leaf | U, S | C | nonseasonal/moderate | ingredient for soup and stir-fried dishes/moderate |
| *Brassica oleracea* L. var. *capitata* L. f. *alba* DC. | Brassicaceae | cabbage | kapisi | biennial herb | leaf | U, S | C | nonseasonal/moderate | used fresh in salads, boiled, stir-fried dishes/moderate |

**Table 2.** *Cont.*

| Scientific Name (VSN [1]) | Family | English Name | Local Name | Life Form | Part Used | L [2] | CC [3] | Seasonality/Availability | Traditional Uses/Cultural Salience |
|---|---|---|---|---|---|---|---|---|---|
| *Capsicum annuum* L. | Solanaceae | bell pepper, chilli | polo pālagi | annual herb | fruit | U, S | C | nonseasonal/moderate | consumed in fresh, dried or processed form as table vegetable or spice/moderate |
| *Cucumis sativus* L. | Cucurbitaceae | cucumber | kukama | annual herb | fruit | U, S | C | nonseasonal/moderate | used for salads/moderate |
| *Cucurbita pepo* L. | Cucurbitaceae | pumpkin, squash | maukeni | annual herb | fruit | U, S | C | nonseasonal/large | baked, boiled and stewed/moderate |
| *Daucus carota* L. | Apiaceae | carrot | karoti | biennial herb | root | U, S | C | nonseasonal/moderate | consumed raw or cooked, alone or in combination with other vegetables, as an ingredient of soups and sauces/moderate |
| *Lagenaria siceraria* (Molina) Standl. | Cucurbitaceae | bottle gourd | pī povi | annual vine | fruit | U | C | nonseasonal/moderate | boiled, fried or in stews/moderate |
| *Lycopersicon esculentum* Mill. | Solanaceae | tomato | tamato | annual herb | fruit | U, S | C | nonseasonal/large | fresh in salads, sauces and as a flavoring ingredient in soups and meat or fish dishes/moderate |
| *Nasturtium officinale* R.Br. | Brassicaceae | watercress | kapisi vai | perennial herb | aerial part | U| | C | nonseasonal/scarce | consumed raw, cooked as spinach or in soups, sometimes as a garnish or condiment/low |
| *Sechium edule* (Jacq.) Sw. | Cucurbitaceae | chayote | soko | perennial vine | fruit | U | C | nonseasonal/moderate | cooked in soups or in stir-fry dishes/moderate |
| *Solanum melongena* L. | Solanaceae | eggplant | isala'elu | perennial herb | fruit | U, S | C | nonseasonal/moderate | roasted, fried, and stuffed/moderate |
| *Vigna unguiculata* (L.) Walp. | Leguminosae | cowpea | pī umi | annual herb | fruit, seed | U, S | C | nonseasonal/moderate | cooked and in soups/moderate |
| Stimulants | | | | | | | | | |
| *Coffea canephora* Pierre ex A.Froehner | Rubiaceae | robusta coffee | kofe | tree | seed | S | C | nonseasonal/moderate | hot watery extract from roasted and ground seeds/moderate |
| *Nicotiana tabacum* L. | Solanaceae | tobacco | tapa'a | annual herb | leaf | U, S | C | nonseasonal/large | cured leaves are smoked/moderate |
| *Piper methysticum* G.Forst. | Piperaceae | kava | ava | shrub | root | U, S | C | nonseasonal/large | ceremonial drink, filtered cold water macerate from pulverized fresh or dried roots/high |
| *Theobroma cacao* L. | Malvaceae | cocoa | koko | tree | seed | U, S | C | seasonal (November-December, April)/large | drunk as cocoa paste added to boiling water sometimes with sugar and milk, eaten when mixed with boiled rice/high |
| Spices | | | | | | | | | |
| *Curcuma longa* L. | Zingiberaceae | turmeric | lega | perennial herb | rhizome | U, S | C | nonseasonal/moderate | food additive and condiment, dye/moderate |
| *Zingiber officinale* Roscoe | Zingiberaceae | ginger | fiu | perennial herb | rhizome | U, S | C | nonseasonal/moderate | food additive and condiment/moderate |
| Algae | | | | | | | | | |
| *Caulerpa racemosa* (Forsskål) J. Agardh | Caulerpaceae | sea grapes | limu fuafua | alga | frond | U | C | nonseasonal/scarce | eaten raw with coconut cream/moderate |

[1] VSN, voucher specimen number, [2] L: Location, S: Savai´i, U: Upolu; [3] CC: Crop category, C: Conventional, N: Neglected.

### 3.3. Market Value and Regional Distribution of Cash Crops

The analysis of economic data collected on plant commodities at local Samoan markets showed that taro was the crop with highest stock volume (25.8%), closely followed by brown and green coconuts (23.8%), pumpkins, squashes and gourds (18.4%), bananas and plantains (13.6%) and papayas (9.6%) and giant taros (5.1%). The stock volumes of all other crops, namely citruses, cucumbers, kava, Samoan cocoa, taro leaf, tobacco and tomatoes, were less than 5%. Pumpkins, squashes and gourds (23.3%), tobacco (17.2%), brown and green coconuts (13.8%), taros (12.6%) and bananas and plantains (11.9%) were crops with the highest stock value. The stock values of citruses, cucumbers, giant taros, kava, papayas, Samoan cocoa, taro leaves and tomatoes were lower than 5%. Tobacco (24.5%) was the crop with the highest stock value per vendor, followed by pumpkins, squashes and gourds (13.0%), brown and green coconuts (8.8%), taros (8.2%), giant taros (8.1%), kava (7.2%), tomatoes (5.9%), Samoan cocoa (5.8%) and bananas and plantains (5.3%). Stock values per vendor for citruses, cucumbers, papayas and taro leaves were less than 5%. The highest number of vendors was selling bananas and plantains (34.1%), pumpkins, squash and gourds (26.9%), brown and green coconuts (23.6%), taro (23.1%), papaya (17.8%), tomato (11.1%), tobacco (10.6%), cucumber (10.1%), Samoan cocoa (8.2%) and giant taro (7.7%). Citruses, kava and taro leaf were sold by less than 5% of vendors. Table 3 provides detailed data on the main cash crops on Samoan markets including product name and category, the number of vendors selling the product, its average price in USD per kg, stock volume in kg, stock value in USD and stock value per vendor in USD. Percentage parameters characterizing economic value of main cash crops on Samoan markets are shown in Figure 1.

**Table 3.** Stock volume and financial value of main cash crops on Samoan markets.

| Product Name (Specification) | Crop Category | Number of Vendors Selling Product (*n* = 208) | Average Price (USD/kg) | Stock | | |
|---|---|---|---|---|---|---|
| | | | | Volume (kg) | Value (USD) | Value per Vendor (USD) |
| Banana and plantain | fruit and nut | 71 | 0.67 | 4558 | 3049 | 43 |
| Brown and green coconut | fruit and nut | 49 | 0.44 | 7997 | 3533 | 72 |
| Citruses | fruit and nut | 6 | 3.43 | 54 | 185 | 31 |
| Cucumber | vegetable | 21 | 1.27 | 524 | 663 | 32 |
| Giant taro | root and tuber | 16 | 0.62 | 1714 | 1058 | 66 |
| Kava (dried powdered root) | stimulant | 8 | 19.57 | 24 | 470 | 59 |
| Papaya | fruit and nut | 37 | 0.33 | 3218 | 1063 | 29 |
| Pumpkin, squash and gourd | vegetable | 56 | 0.96 | 6180 | 5963 | 106 |
| Samoan cocoa (dried cocoa paste) | stimulant | 17 | 9.04 | 88 | 796 | 47 |
| Taro | root and tuber | 48 | 0.37 | 8670 | 3235 | 67 |
| Taro leaf | vegetable | 7 | 1.66 | 63 | 104 | 15 |
| Tobacco | stimulant | 22 | 30.38 | 145 | 4404 | 200 |
| Tomato | vegetable | 23 | 3.23 | 339 | 1093 | 48 |

Tuamasaga is the most important district for Samoan market supply because of the large financial value of its production (44.8%), the highest number of vendors (43.2%) and the greatest product diversity (20.6%). It is followed by Gagaemauga district, where 25.3% of the production financial value, 14.9% of vendors and 15.9% of product diversity originate. In Savai´i, Palauli supplies the local market with the highest product diversity (25.0%), whereas Faasaleleaga generates the highest financial value (29.7%). Faasaleleaga and Gagaemauga are the districts accommodating the most vendors in Savai´i (33.3% and 31.4% respectively). Tuamasaga is the most important region in all three parameters in Upolu (56.4% of production financial value, 64.9% of vendors and 31.4% of product diversity). With a majority of production financial value (78.9%), vendors (65.5%) and product diversity (55.6%), Upolu is a significantly more important market than Savai´i. Although exchange between local markets on both islands is generally low, more production (1.6%), vendors (1.4%) and crop products (7.9%) move from Savai´i to Upolu than in the opposite direction (0.3%, 0.7% and 3.2%, respectively). A map illustrating the quantified distribution of financial values of market production, the number of vendors and crop diversity among individual districts is shown in Figure 2.

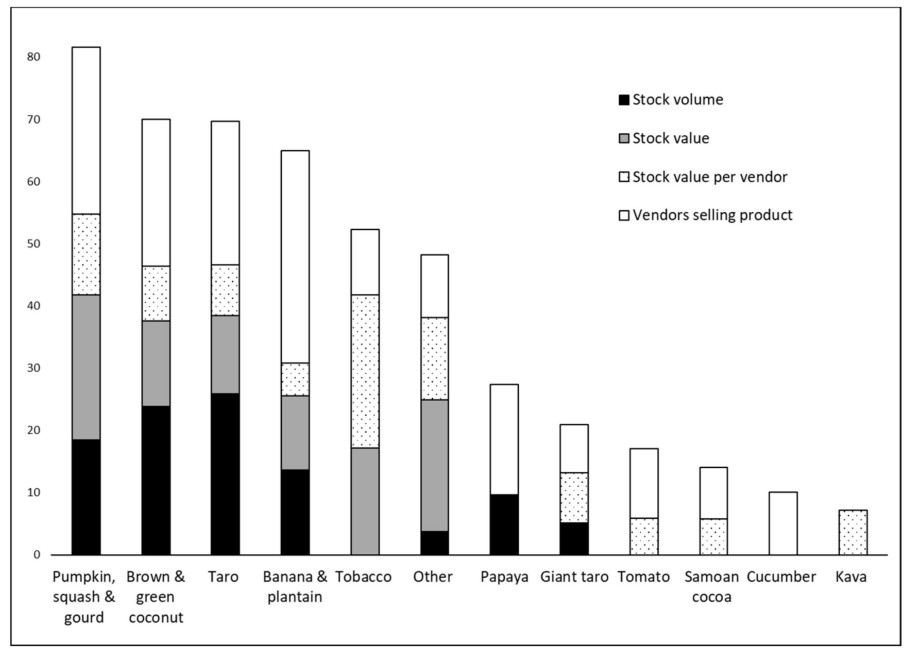

**Figure 1.** Parameters characterizing economic value of main cash crops on Samoan markets (in%).

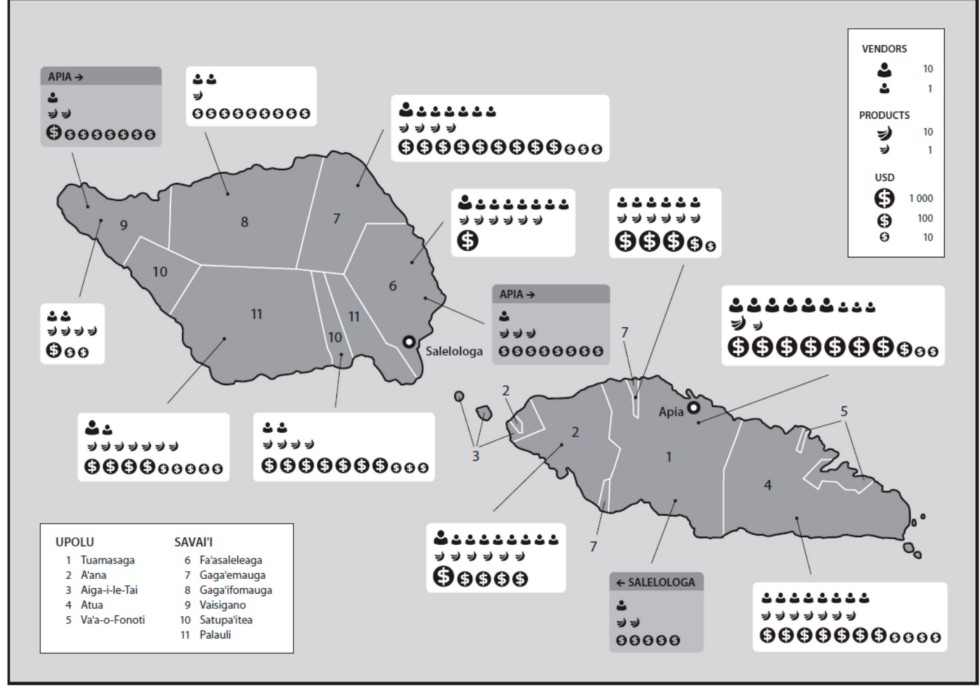

**Figure 2.** Regional distribution of vendors, diversity and market value of cash crops at Samoan markets.

## 4. Discussion

Based on the analysis of data collected during our survey, a typical vendor at the local Samoan market can be described as a female between 31 and 60 years of age living in Upolu. These demographic characteristics correspond well with those observed in previous studies focused on the inventory of various plant commodities sold on local markets in developing regions. For example, Bussmann et al. [36] reported that vendors selling medicinal plants in the markets of La Paz and El Alto in Bolivia were women of an estimated 25–70 years of age. Similarly, women were the main vendors at the wild

edible plant markets in Kisangani in the Democratic Republic of Congo [24] and at traditional markets of medicinal plants in Southern Ecuador [37].

As far as diversity of plants commercialized in the local Samoan markets is concerned, the assortment identified during our survey is similar to that described by Whistler [13]. In the period 1995–1996, he recorded 61 various items of fruits, vegetables, starch and miscellaneous crops sold in the market places in Savai´i, Upolu and Tutuila (American Samoa). Although most species are available year round, some products are seasonal and highly perishable (e.g., fruits of *Artocarpus.. altilis*). Unexpected changes in yield and seasonality of these crops could cause significant economic losses, wasted resources and disrupt local food supplies. This may be mitigated through careful selection of cultivars to extend the season or to help plan for processing them into more stable products [38]. Most of the species identified during our study in the market were common agricultural crops (e.g., *Cocos nucifera*, *Colocasia esculenta*, *Cucurbita pepo* and *M. × paradisiaca*), however, less known food plants, namely *A. pavonina*, *I. fagifer*, *M. troglodytarum* and *T. leontopetaloides*, were also present. It has been experimentally proven that plants that are used traditionally as foods but have not been adopted by large-scale agriculture (known also as neglected and underutilized crops) may contain higher amounts of specific health-beneficial constituents (e.g., vitamins, minerals, fiber, secondary metabolites and fatty and amino acids) than conventional agricultural crops. Due to their nutritional and medicinal values, these species could highly contribute to the improvement of both the nutritional and health status of the local populations in rural areas of developing countries and, as a result, increase sustainable food and nutrition security in low-income regions. For example, *M. troglodytarum* that was found to contain significant levels of β-carotene has been suggested as prospective food for vitamin A deficiency and chronic disease prevention programs in the Pacific [39]. A local variety of banana (*M. × paradisiaca*; group AAB), called *fa´i sāmoa*, can be another interesting material for future research based on an analysis of its nutritional properties. Recently, our team proposed seeds of *A. pavonina* and *I. fagifer* as promising sources of specific nutrients and compounds, such as fatty acids, minerals, phenolics and vitamins, with potential to reduce the risks of overweightness and obesity-related diseases in Samoa. These underutilized crops have also been proposed for development of novel foods with enhanced health-related properties [40,41]. In contrast to above mentioned species, the nutritional value of *T. leontopetaloides*, a neglected crop producing starchy tubers used as an additive in traditional Samoan foods [13], is described in general terms only [42]. Since *I. fagifer* and *T. leontopetaloides* are naturally occurring in the littoral zone and coastal swamp forests [13], they are better adapted to local environments, including regions with abiotic stress conditions (e.g., high salinity soils), than most conventional crops. Thus, both these underutilized species provide a better chance for organic and low-input agriculture and their growing will have less negative impact on the environment and local ecosystems.

Since agriculture sales belong to the most important sources of household incomes in Samoa [43], the monitoring of common economic indicators such as price, stock volume and the financial value of cash crops can provide useful information on the overall strength of the local agricultural market and general economic health of the country. Since 2008, SBS monitors the prices and volumes of selected agricultural produce at the local markets every month. These products include fruits and nuts (banana, breadfruit and coconut), vegetables (cabbage, cucumber, pumpkin and tomato) and root and tuber crops (tannia, taro and yam) [44]. The results of our survey correspond well with SBS data and with the report of Tamasese [45]. However, they newly indicate that the stimulants (kava, Samoan cocoa and tobacco), along with papaya are also commodities of market interest. Although the contribution margin for agricultural products is small in a typical developing market economy, processes that add value to them can increase their market potential [46]. In Samoa, the food sector actors suggested that local foods could create opportunities for economic growth, particularly through their increased production for export [47]. Samoan cocoa (known as a koko sāmoa), a popular Samoan drink made from roasted cocoa seeds crushed into a paste and sold in the shape of a dried ball or cup [13], can be mentioned as a typical example of a local food product with added value. Similarly, the dried fruits

of *Carica papaya* have been proposed as a lucrative value-added agricultural product in the local Samoan market and for export to New Zealand [46]. Therefore, both these commodities represent a good opportunity for Samoan farmers, market vendors and food producers to earn more income. Concerning tobacco, it is well known that smoking it is an important health risk factor contributing to the dramatic increase in the worldwide diagnosis of cancer, and of cardiovascular and chronic respiratory diseases [48,49]. The incidence of all of these health complaints has increased significantly in Samoa recently [50]. According to our results, the local markets are important sources of tobacco for the Samoan population, which suggests one question the appropriateness and effectiveness of current strategies for the regulation and control of local trade in this commodity. *Piper methysticum*, a stimulant originating in Polynesia, is an important Samoan domestic and export cash crop, grown by subsistence farmers and larger commercial growers. It is exported to the Pacific Rim markets of Australia, New Zealand and the USA and to neighboring countries such as Fiji. It has been reported that most kava in Samoa is grown on Savai´i [51], which is supported by our survey results showing a relatively high abundance of this commodity in Savai´i market. Especially in this region where society is still following the traditional way of life, local products such as kava help also social sustainability because they favor contacts between producers and consumers, which influences greater information and knowledge of what is consumed and enables developing a feeling of community [52]. The drinking of ceremonial beverage made from the root of the *P. methysticum* is one of the most important customs of the Samoa Islands, which takes place on most important occasions including the bestowal of a chiefly title, formal occasions and events, significant gatherings and meetings or welcoming and bidding farewell to guests and visitors. Another cultural tradition is the practice of food preparation called umu. This is a local type of above ground earth oven of hot volcanic stones. The Samoan umu starts with a fire to heat rocks, which are then stacked around parcels of food wrapped in banana, breadfruit or taro leaves. Pork, fish, chicken and many traditional plant foods, including taro and breadfruit, are all cooked at the same time without their flavors mingling. Coconut cream is an essential part of many umu recipes [13].

It is well known that the production of crops in the most suitable regions, combined with the ability to transport crop commodities from one region to another, contributes to the comparative advantage of a country. Therefore, a detailed knowledge of the geographical distribution of a particular food commodity in a specific area may harbor promising characteristics for a transition to sustainable food systems in the targeted region [53]. Our results suggest that exchange of crop commodities between local markets in both islands is generally low. This finding can be supported by the observations of Hardin and Ting Kwauk [47] who have previously reported that shipping and transportation have previously been identified as common challenges to the supplying of the Samoan market with affordable foods. Similarly, limited and costly transport options have been identified as serious barriers for access to local markets for subsistence farmers in Fiji, especially for those from remote communities [54]. Although Underhill et al. [55] have previously reported that transport logistics do not have a negative impact on fruit and vegetable markets in Samoa, our results showing the irregular distribution of crop production areas among the islands and their districts suggest that an optimization of the logistics of marketed crop commodities can contribute to an increase in the efficiency of the local agricultural sector. In addition, the improvement of logistics and supply chain management can reduce the negative effect of transportation and storage on nutritional quality and sensory properties of local food products [56]. Providing healthier and better tasting food will subsequently improve the quality of life of Samoans. Moreover, decreased pollution from transportation will minimize negative impacts on the environment, health and well-being of local communities [57]. This will contribute to the environmental, social and health sustainability of the region and its population.

## 5. Conclusions

In conclusion, our results suggest that assortment and economical value of plant food products at local markets in developing countries have the potential to increase sustainable food and nutrition

security of the local population and support economic growth of these regions. For example, neglected/underutilized crops available at local markets in Samoa such as *A. pavonina*, *I. fagifer*, *M. troglodytarum* and *T. leontopetaloides* have the potential for development of new food products with beneficial nutritional and health properties and for dietary diversification and provision of micronutrients to local population. Moreover, certain commodities available in local markets in larger quantities (e.g., papaya) or having higher added economic value (e.g., kava and Samoan cocoa) seem to have promising potential for export. In addition, our findings suggest that development of appropriate processing technologies and the optimization of the logistics of crop products sold at markets in developing countries can contribute to an increase in efficiency of the local agricultural sector.

**Author Contributions:** Economic analysis of collected data and manuscript drafting, V.V.; market inventory and analysis of ethnobotanical data, P.N.; participation in data analysis and manuscript preparation, J.T.; participation in the inventory of local markets, L.H.; verification of the ethnobotanical data and participation in their analysis, J.W.S.; collection of the plant materials and processing the voucher specimens, T.K.; participation in the collection of market data and plant materials. F.L.; conceptualization and coordination of the whole study, botanical identification of crop species, and finalization of the manuscript. L.K. All authors have read and agreed to the published version of the manuscript.

**Funding:** This research was funded by Internal Grant Agency of the Faculty of Tropical AgriSciences, grant number IGA 20205001.

**Acknowledgments:** The authors are grateful to Micheal Ua Seaghdha for English linguistic revision of the manuscript.

**Conflicts of Interest:** The authors declare no conflict of interest.

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
