# Peer review of "Diversity, Economic Value and Regional Distribution of Plant Food Products at Local Tropical Markets: A Samoa Case Study"

_sustainability, doi:10.3390/su122310014_

Round 1

Reviewer 1 Report

Well written and argument article. I do miss the overview of the legal framework of marketing plant products on the markets of Samoa (for ideas see https://www.sciencedirect.com/science/article/pii/S2405844020300670

Also, given that the authors have done they research all over the year, it would be important to show in Table 2 the seasonal availability of the plant products and ad some discussion on the seasonal food security. 

The information provided on page 5, rows 193-210 would be better to present in some form of a diagram, would be easier for the reader to grasp. 

Page numeration is messed up, Table 2 starts the new numeration and after it comes the next start. 

On row 231 is an abundant word "Authors"

Reviewer 2 Report

The paper has a good potential, but if authors want seriously to write about the opportunities that local food products for the food sustainability authors NEED to present more data and related discussion on the 1. environmental/ecological availability/sustainability of these plants and on the 2. CULTURAL VALUE of the recorded plants - CULTURAL VALUES means to me here local perceptions and food uses.

Without these details, the paper in its actual state is just a mere list of sold crops. 

Additionally, 3. the authors should discuss more the social articulation and social values attached to these markets and marketed plants and therefore their significance in terms of SOCIAL sustainability.

Reviewer 3 Report

The authors of the manuscript entitled "Diversity, economic value and regional distribution of plant food products at local tropical markets: A Samoa case study" collected and analyzed data on the range and economic value of plant food that have the potential to increase Samoa's food supply sustainability.

The formulated objectives of the study were defined and discussed using appropriate methods. The plant-food species available in the local Samoa markets have been documented. Moreover, the sales volume and the value of the products were determined. Finally, the possibilities of food distribution on the described islands of Upolu and Savai'i were mapped.

The article is a valuable source of information on the growing economy of Samoa.

In my opinion, the article can be accepted in the present form.

Round 2

Reviewer 2 Report

Authors have de-facto ignored the simple points I raised - which are pretty  basic in economic botany or ethnobotany.

Let's be clearer:

1 Authors HAVE TO include in the table a column about seasonality (months) and availability (large moderate scarce) of the plant items recorded in the visited markets;

2 Authors HAVE TO include in the table the cultural importance of the items - ie. A traditional detailed uses; B cultural salience (high moderate low)

3. Authors NEEDS to WELL articulate the meaning of their additional 1. and 2. in the discussion - not just to add single no-sense phrases about rituals. 

Author Response

Itemized response to the comments of reviewer

Query: Authors HAVE TO include in the table a column about seasonality (months) and availability (large moderate scarce) of the plant items recorded in the visited markets.

Response: we included in the Table 2 a column about seasonality and availability of the crops recorded in the Samoan markets.

Query: Authors HAVE TO include in the table the cultural importance of the items – i.e. A traditional detailed uses; B cultural salience (high moderate low).

Response: we included in the Table 2 the traditional detailed uses and cultural salience of the crops recorded in the Samoan markets.

Query: Authors NEEDS to WELL articulate the meaning of their additional 1. and 2. in the discussion - not just to add single no-sense phrases about rituals.

Response: Although the main aim of the research was to identify economic value of crops on Samoan markets, with aim to discuss their traditional use we added following texts to the end of third paragraph of the Discussion section:

Another cultural tradition is the practice of food preparation called umu. This is a local type of above ground earth oven of hot volcanic stones. The Samoan umu starts with a fire to heat rocks, which are then stacked around parcels of food wrapped in banana, breadfruit or taro leaves. Pork, fish, chicken and many traditional plant foods, including taro and breadfruit, are all cooked at the same time without their flavours mingling. Coconut cream is an essential part of many umu recipes [13].

In addition, we added following text to the end of the Diversity of commercialized plants subsection of the Results section:

Nonseasonal, seasonal and imported (not cultivated in Samoa) crops accounted for 77.4 %, 20.8 % and 1.9 % of total number of plant species sold in the markets, respectively. The most crops were of moderate availability (58.5%) and cultural salience (69.8 %). Largely available (22.6 %) and highly culturally important (18.9) plants were dominant over the scarce (18.9 %) and low salient (11.3 %) species. The detailed data on plant species identified in Samoan markets including their category of use, life form, part used, locality, crop category, family, seasonality, availability, traditional uses and cultural salience as well as their scientific, English, and local names are listed in Table 2.

Round 3

Reviewer 2 Report

The authors successfully answered the points I raised this time and I am happy with their revision.